# Pregnancy Outcome, Antibodies, and Placental Pathology in SARS-CoV-2 Infection during Early Pregnancy

**DOI:** 10.3390/ijerph18115709

**Published:** 2021-05-26

**Authors:** Won-Kyu Jang, Su-Yeon Lee, Sunggyun Park, Nam Hee Ryoo, Ilseon Hwang, Ji Min Park, Jin-Gon Bae

**Affiliations:** 1Department of Obstetrics and Gynecology, Keimyung University School of Medicine, Dongsan Medical Center, Daegu 42601, Korea; cindeln@naver.com (W.-K.J.); feedback54@naver.com (S.-Y.L.); 2Departments of Laboratory Medicine, Keimyung University School of Medicine, Dongsan Medical Center, Daegu 42601, Korea; nosdolu@dsmc.or.kr (S.P.); nhryoo@dsmc.or.kr (N.H.R.); 3Department of Pathology, Keimyung University School of Medicine, Dongsan Medical Center, Daegu 42601, Korea; ilseon@dsmc.or.kr (I.H.); Park starnmin28@gmail.com (J.M.P.)

**Keywords:** COVID-19, SARS-CoV-2, vertical infection transmission, COVID-19 antibody, neutralizing antibody, first trimester pregnancy, placenta, pathology

## Abstract

There are reports that pregnant women infected with SARS-CoV-2 not only have increased morbidity but also increased complications and evidence of maternal and fetal vascular malperfusion on placental pathology. This was a retrospective study of pregnant women diagnosed with SARS-CoV-2 infection after March 2020. The results of reverse transcription polymerase chain reaction testing and IgM and IgG antibody testing of the amniotic fluid, cord blood, placenta, and maternal blood were confirmed at delivery. Placentas were evaluated histopathologically. The study included seven pregnant women diagnosed with SARS-CoV-2 infection during pregnancy at a mean gestational age of 14.5 weeks. Out of the seven women, five were infected during the first trimester. The mean gestational age at delivery was 38.4 weeks. The reverse transcription polymerase chain reaction results for maternal plasma, cord blood, placenta, and amniotic fluid were negative and IgG antibodies were detected in maternal plasma and cord blood. On placental pathology, maternal vascular malperfusion was found in only one case, fetal vascular malperfusion in four cases, and inflammatory changes were found in two cases. Pregnancy outcomes for women diagnosed with SARS-CoV-2 infection during early pregnancy are positive and it is likely that maternal antibodies are passed to the fetus, which results in a period of immunity.

## 1. Introduction

In December 2019, Severe Acute Respiratory Distress Syndrome Coronavirus 2 (SARS-CoV-2), a novel coronavirus that has both toxic and infectious properties, first appeared in Wuhan, China. On 1 April 2021, there were more than 100 million infections worldwide and more than 2 million deaths [1,2]. This count is increasing daily and the approximate worldwide mortality rate is estimated at 2–3% as reported by the World Health Organization [2].

The coronavirus disease 2019 (COVID-19) occurs with infection by SARS-CoV-2. The infection may be asymptomatic, result in a mild viral syndrome, or cause severe symptoms such as respiratory failure, shock, multi-organ dysfunction, and death [3].

Previous studies of Middle East Respiratory Syndrome Coronavirus (MERS-CoV) and SARS-CoV indicate that infections during pregnancy are severe and tend to be associated with poor neonatal outcomes including an increased risk of miscarriage, fetal growth restriction, and preterm birth [4,5,6,7]. However, most of the published studies documenting these results involved infection during the second or third trimester of pregnancy and there were only a few cases reported of first trimester infection where the pregnancy continued to a term birth [8].

Based on these findings, we reviewed seven cases that occurred at our hospital. We were able to determine the results of cord blood antibody and placental pathology, which to our knowledge have not been previously reported, as well as the pregnancy outcomes of cases with first trimester infection that continued to delivery.

## 2. Materials and Methods

Following approval from the institutional review board, we conducted a retrospective chart review of pregnant women infected with COVID-19 during pregnancy who delivered at Keimyung University Dongsan Medical Center, Daegu, Korea after March 2020.

The diagnosis of COVID-19 was confirmed by real-time reverse transcriptase polymerase chain reaction (real-time RT-PCR) using Allplex 2019-nCoV (Seegene, Seoul, Korea) as per the manufacturer’s instructions with a nasopharyngeal swab. All RNA was extracted from the swab using the Real Prep Automated Nucleic Acid Extraction System (Biosewoom, Seoul, Korea). CFX96 (Bio-Rad Laboratories, Hercules, CA, USA) was used as a thermocycler for the real-time RT-PCR. Cycle threshold (Ct) values from FAM (E gene), Cal Red 610 (RDRP gene), Quasar 670 (N gene), and HEX (internal control) were acquired. We used 40 as the cut-off value for Ct in all target genes. Mothers confirmed with COVID-19 infection were isolated and observed for symptoms and blood tests and chest radiographs were performed. Two negative real-time RT-PCR tests were considered to be evidence of recovery from COVID-19 and maternal quarantine was lifted at that point. Following delivery, newborns were immediately moved to a negative pressure ward if necessary, quarantined, and real-time RT-PCR was performed with a nasopharyngeal swab. At the time of delivery, cord blood, maternal blood, amniotic fluid, and placental specimens were collected for evidence of infection.

We reviewed the mother’s medical records, clinical characteristics, blood test results, placental biopsies, and neonatal outcomes. The RNA was extracted from these samples (amniotic fluid, placenta, cord blood, maternal blood, and nasopharyngeal swab) using the same method outlined above. Serum IgM and IgG antibodies against SARS-CoV-2 were measured using the immunochromatographic test kit STANDARD Q COVID-19 IgM/IgG Plus Test (SD biosensor, Suwon, Korea) according to the manufacturer’s instructions. The Elecsys Anti-SARS-CoV-2 assay (Roche Diagnostics, Rotkreuz, Switzerland) was performed according to the manufacturer’s instructions on Cobas e801 (Roche Diagnostics, Rotkreuz, Switzerland) for detecting total antibody against SARS-CoV-2. Quantitative results were obtained and a cut-off index (COI) of 1 was used for qualitative results. Placental specimens were evaluated and reviewed histopathologically by pathologists.

Statistical analyses were performed using the Statistical Package for the Social Sciences (SPSS 25.0, IBM Corporation, Armonk, NY, USA). Variables were expressed as the mean and number and descriptive statistics were used as well.

## 3. Results

The study included seven pregnant women diagnosed with COVID-19 during pregnancy with a mean gestational age of 14.5 weeks. Out of these, five (71.4%) were infected during the first trimester. All seven patients had no underlying diseases and no preterm birth history (Table 1). Symptoms such as myalgia, cough, sore throat, sputum production, and rhinorrhea were all mild except for case number 3, who had fever and dyspnea (Table 2).

The mean gestational age at delivery was 38.4 weeks, along with a mean time of 166.4 days from infection onset to delivery. The mean birth weight was 3.1 kg. A total of 60% of the deliveries were delivered by C-section. Only one infant was born late preterm at 36 weeks and 4 days. There was only one NICU admission and that was the infant born late preterm. He was admitted to the NICU for transient tachypnea, which resolved quickly, and he was discharged home with the mother (Table 3).

Real-time RT-PCR results from maternal plasma, cord blood, placenta, and amniotic fluid were all negative. Amniotic fluid could not be collected due to vaginal delivery in one case. IgM antibody was found in two cases and IgG antibody was found in four other cases in maternal plasma collected at the time of delivery. The IgM antibody was not detected in cord blood and IgG antibodies were found in all but one sample of cord blood. Antibody titers were all positive in both maternal plasma and cord blood (Table 4).

Pathologic features of maternal vascular malperfusion included excessive villous infarction (thrombosis of stem villi, endothelial injury, thickened walls of stem villous arterioles, and luminal obliteration), increased fibrin deposition, accelerated villous maturation (increased syncytial knots defined as knots in >20% of villi preterm or 30% of villi at term), increased calcification, and intervillous thrombosis. Pathologic features of fetal vascular malperfusion included extensive avascular villi (groups of homogeneously pink villi may contain stromal cells but lack capillaries), karyorrhexis (fragmented red blood cells and cellular debris over fetal vessels), chorangiosis (more than 10 fields of placental parenchyma with more than 10 terminal villi, each containing more than 10 capillaries in at least three different areas), distal villous hypoplasia (paucity of distal villi in relation to surrounding stem villi seen in lower 2/3 of parenchymal thickness, involving at least 30% of full thickness), and delayed villous maturation. Pathologic features of inflammatory changes included chronic villitis (infiltrate of lymphocytes with histiocytes), diffuse villous edema (enlarged villi, loose mesenchymal-appearing stroma, and central thin-walled vessels), and chorioamnionitis/subchorionitis. Chorioamnionitis/subchorionitis was classified into stage 1 (acute inflammation limited to subchorionic space or membranous trophoblast, not extending into fibrous chorion), stage 2 (acute inflammation of fibrous chorion/amnion), and stage 3 (necrotizing acute inflammation with neutrophil karyorrhexis and amniocyte necrosis); and grade 1 (scattered neutrophils) and grade 2 (confluent neutrophils with more than 200 cells in extent). In one case (case 7), the placenta showed increased fibrin deposition and increased calcification, which are features of maternal malperfusion. Chorangiosis and distal villous hypoplasia, features of fetal malperfusion, were present in cases 4 and 5 and cases 2 and 6, respectively. Features of chronic inflammatory changes were observed in two cases as chorioamnionitis/subchorionitis stage 1 and grade 1 (cases 1 and 7, respectively) (Table 5 and Figure 1).

## 4. Discussion

The results of this study show that the pregnancy outcomes of women diagnosed with COVID-19 during early pregnancy are positive. In particular, in the case of mothers infected during the first trimester, there were no fetal or neonatal deaths and the newborns did not have any infectious symptoms. There appeared to be no vertical transmission of the virus. Maternal IgG antibody generated after infection appeared to be transmitted to the fetus.

Currently, there is limited information on the effects of COVID-19 infection during pregnancy on maternal, fetal, placental, and neonatal outcomes. In recent review articles, the majority of COVID-19 infections during pregnancy occurred during the third trimester, most of the rest occured during the second trimester, and very few infections occurred during the first trimester [9,10]. In this study, five cases of first trimester infections were reported, which is currently the most in the published literature. This study found that most of the infections possess mild clinical manifestations with positive pregnancy outcomes and no fetal or neonatal deaths. This study found very positive pregnancy outcomes compared with the results of previous reports of influenza virus, SARS-CoV, MERS-CoV, and reports on abortion and severe adverse pregnancy outcomes of COVID-19. This is likely due to the fact that most of the patient’s symptoms were mild. In fact, the one case with fever and chest discomfort delivered prematurely following preterm premature rupture of fetal membranes.

In this study, IgG antibody was negative in maternal plasma and cord blood in only one case. However, since the antibody titer was positive, it was considered to be a false negative resulting in IgG antibody being detected in all cases. As noted in previous studies, when IgG antibodies are observed, neutralizing antibodies are simultaneously found to be present and maternal antibodies to specific pathogens can be delivered through the placenta to the fetus to protect the newborn from infection [11]. A prior report noted that the correlation between IgG antibody and neutralizing antibody is similar in SARS-CoV-2 [12]. This suggests that the newborn has both IgG and neutralizing antibody delivered from the mother and the newborn is likely to possess some immunity to COVID-19 immediately following delivery.

Shanes et al. reported that the placentas of mothers infected with COVID-19 had at least one feature of maternal vascular malperfusion compared to a control group [13]. Similarly, in a study published by Menter et al., maternal vascular malperfusion was observed in all five cases [14]. However, in this study, maternal vascular malperfusion was found in only one of seven cases. The main cause of maternal vascular malperfusion in placental pathology is probably due to the increased vortex or underperfusion in maternal placental circulation as a result of shock and hypoxia. This maternal vascular malperfusion is well described in maternal placentas diagnosed with preeclampsia or fetal growth restriction during pregnancy [15]. In fact, case 7 in our study was delivered at 39 weeks and 3 days of gestation, but the neonate birth weight was 2.51 kg, which was less than 10th percentile based on the week of birth, indicating fetal growth restriction; this is thought to have resulted from maternal vascular malperfusion. The mechanism of maternal vascular malperfusion in patients infected with SARS-CoV-2 still requires further study, but intervillous thrombi increases with COVID-19, which leads to an increase in the vortex or underperfusion in maternal placental circulation, thereby causing maternal vascular malperfusion [13,14]. However, in previous studies, this maternal vascular malperfusion was not observed, even in cases of SARS infection during the first trimester [16]. Presumably, in the first trimester, SARS infection had an all-or-nothing effect and maternal vascular malperfusion was not observed if pregnancy complications do not develop after this period. The cases in this paper demonstrate the longest intervals in gestational age with an ongoing SARS-CoV-2 infection, that is, from the time of infection to delivery. Because of SARS-CoV-2 infection in the early trimester, especially in the first trimester, this case of maternal vascular malperfusion was not seen or thought to have recovered at the time of delivery. Five out of seven cases of infection in this study occurred during the first trimester. The symptoms at the time of SARS-CoV-2 infection were mild, no specific pregnancy complications developed, and maternal vascular malperfusion was not observed. The fetal vascular malperfusion findings in this study were not significantly different from those of the previous two studies [13,14]. In this study, placental inflammation was observed in two cases. In both cases, chorioamnionitis/subchorionitis stage 1 and grade 1 were observed but clinically no findings were found that indicated inflammation except for SARS-CoV-2 infection during pregnancy. These findings indicate the possibility of sequelae from SARS-CoV-2 infection during early pregnancy.

The main limitations of this study are its retrospective nature and small sample size. Since it is a retrospective study, there is no direct data on the infants immediately following birth which limits the study of antibodies in these infants. The advantages of this study include data collection from COVID-19 infected mothers beginning from the first trimester, confirmation of antibody transmission to the fetus, and placental biopsy in COVID-19 infected mothers, which has not been previously well reported.

## 5. Conclusions

In conclusion, the pregnancy outcome in women diagnosed with COVID-19 infection during the first trimester appears to be positive and maternal antibodies are passed through the placenta to the fetus; thus, the newborn is likely to be born with some immunity. Evidence of fetal vascular malperfusion is increased on placental pathology.

## Figures and Tables

**Figure 1 ijerph-18-05709-f001:**
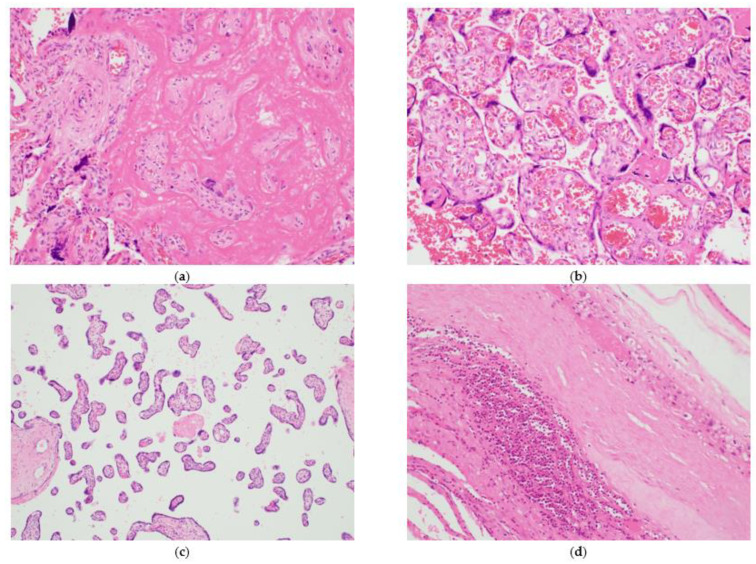
Pathologic features of placentas: (**a**) fibrin deposition in patient 7: the section is characterized by the extensive deposition of fibrinoid material in the intervillous space (200×); (**b**) chorangiosis in patient 4: chorionic villi showing an increased number of capillaries with a dilated lumen (200×); (**c**) distal villous hypoplasia in patients 2 and 7: the section reveals long, elongated, and edematous villi with reduced numbers of syncytial knots (100×); (**d**) chorioamnionitis in patients 1 and 7: neutrophilic infiltrate of chorion was present and amnion as intact (200×).

**Table 1 ijerph-18-05709-t001:** Clinical characteristics.

Case Number	1	2	3	4	5	6	7
Maternal age (years)	40	33	35	36	22	39	37
BMI	30.4	23.8	20.9	24.9	29.0	24.2	19.9
Gravida	3	2	3	3	1	2	3
Para	1	1	1	2	0	1	2
History of preterm birth	0	0	0	0	0	0	0
Previous illness	No	No	No	No	No	No	No
GA at infection	30.4	24.5	13.6	11	8.2	7.3	6.5
Time to negative results (days)	47	19	19	13	52	10	28

BMI, body mass index; GA, gestational age.

**Table 2 ijerph-18-05709-t002:** Signs and symptoms of disease.

Case Number	1	2	3	4	5	6	7
Fever	No	No	Yes	No	No	No	No
Myalgia	Yes	No	Yes	No	Yes	No	Yes
Cough	Yes	No	Yes	Yes	No	Yes	No
Dyspnea	No	No	Yes	No	No	No	No
Sore throat	Yes	No	Yes	No	No	No	Yes
Sputum production	Yes	Yes	Yes	No	No	Yes	No
Rhinorrhea	No	Yes	No	No	No	No	No

**Table 3 ijerph-18-05709-t003:** Neonatal outcomes.

Case Number	1	2	3	4	5	6	7
GA at delivery (weeks)	38.5	38	36.4	39	38.1	40	39.3
Onset of infection to delivery (days)	57	93	159	196	203	228	229
Delivery mode	CS	CS	CS	VD	CS	VD	VD
Birth weight (kg)	3.67	2.90	2.40	3.55	3.32	3.15	2.51
Apgar score 1/5 min	8/9	8/9	7/8	8/9	8/9	8/9	8/9
Sex	F	F	M	F	M	M	F
NICU admission	No	No	Yes *	No	No	No	No
Anomalies	No	No	No	No	No	No	No

GA, gestational age; CS, cesarean section; VD, vaginal delivery; F, female; M, male. * Hospitalized for symptoms of transient tachypnea.

**Table 4 ijerph-18-05709-t004:** Laboratory findings at delivery.

Case Number	1	2	3	4	5	6	7
Mother’s Plasma PCR	Neg	Neg	Neg	Neg	Neg	Neg	Neg
Cord blood PCR	Neg	Neg	Neg	Neg	Neg	Neg	Neg
Placental PCR	Neg	Neg	Neg	Neg	Neg	Neg	Neg
Amniotic fluid PCR	X *	Neg	Neg	Neg	Neg	Neg	Neg
IgM antibody in maternal plasma	Yes	No	No	No	No	No	Yes
IgG antibody in maternal plasma	Yes	Yes	Yes	No	Yes	Yes	Yes
IgM antibody in cord blood	No	No	No	No	No	No	No
IgG antibody in cord blood	Yes	Yes	Yes	No	Yes	Yes	Yes
COI of total anti-SARS-CoV-2 antibody in Maternal plasma	77.6	5.4	6.79	6.29	13.4	1.1	3.17
COI of total anti-SARS-CoV-2 antibody in Cord blood	71.5	5.42	8.15	12.3	17.5	1.63	5.84

COI, cut-off index. * Failure to collect amniotic fluid.

**Table 5 ijerph-18-05709-t005:** Placental pathologic findings.

Case number	1	2	3	4	5	6	7
Maternal vascular malperfusion							
Excessive villous infarction	No	No	No	No	No	No	No
Increased fibrin deposition	No	No	No	No	No	No	Yes
Accelerated villous maturation	No	No	No	No	No	No	No
Increased calcification	No	No	No	No	No	No	Yes
Intervillous thrombosis	No	No	No	No	No	No	No
Fetal vascular malperfusion							
Extensive avascular villi	No	No	No	No	No	No	No
Karyorrhexis	No	No	No	No	No	No	No
Chorangiosis	No	No	No	Yes	Yes	No	No
Distal villous hypoplasia	No	Yes	No	No	No	Yes	No
Delayed villous maturation	No	No	No	No	No	No	No
Inflammatory change							
Chronic villitis	No	No	No	No	No	No	No
Diffuse villous edema	No	No	No	No	No	No	No
Chorioamnionitis/subchorionitis	Yes	No	No	No	No	No	Yes

## Data Availability

Data sharing is not applicable to this article.

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
