# Peer review of "Pregnancy Outcome, Antibodies, and Placental Pathology in SARS-CoV-2 Infection during Early Pregnancy"

_ijerph, 2021, doi:10.3390/ijerph18115709_

Round 1
Reviewer 1 Report
In this retrospective study, the authors assessed pregnancy outcomes with a number of parameters in women diagnosed with Covid-19 during the first trimester. This reviewer had several concerns that need to be addressed before publication.
- It is not clear how the authors diagnosed the pathological features of maternal vascular malperfusion and fetal vascular malperfusion, including excessive villous infarction, accelerated villous maturation, extensive avascular villi, distal villous hypoplasia, delayed villous maturation, chronic villitis, etc. What are their criteria? Were they diagnosed by pathologists?
- Can the authors provide the images of these placental tissues showing those pathological alterations?
- Can the authors explain why maternal vascular malperfusion occurred in women with shorter interval from infection with the virus to delivery as shown in Shanes et al but did not happen in the current study with longer interval?
Reviewer 2 Report
I agree with the authors that information about SARS-CoV-2 infections during pregnancy is important.
However, the extremely small "n" cannot compete with the previously published population and cohort studies that have described the impact of SARS-CoV-2 infections in pregnant women (Crovetto, Clin Infect Dis, 2021) (Adhikari, JAMA, 2020). I do not think that this manuscript provides novel information pertaining to Covid-19 and pregnancy outcomes.
Reviewer 3 Report
This is a retrospective study of pregnant women diagnosed with SARS-COV2 infection after March 2020. It shows that the outcome of these women during early pregnancy is good. There was no neonatal deaths and no vertical trasmission of the virus. As the authors admit, the major limitation of the study is the small numbers of women tested.
Question 1: How did you measure the placenta inflammation? By which mecchanisms do you think it may be caused? How could you explain it was just related to 2 causes? Do these mothers present similar clinical manifestations?
Question 2: Was it possible to follow the antibodies development in the new born? Was it possible to perform a neonatal pharyngeal swab sample?
Question 3: Apart from IgG and IgM, did the author look the level of interferon in the pregnant women and in the children? This is not mention, although we know that, especially in mild form of coronavirus disease, we do have an increase of type I IFN.
